# The impact of socio-economic institutional change on agricultural carbon dioxide emission reduction in China

**Deng Jie Long** [ID]*, **Li Tang**

School of Marxism, Hefei University of Technology, Hefei, China

* long_bnm@126.com

## Abstract

With the change of social economic system and the rapid growth of agricultural economy in China, the amount of agricultural energy consumption and carbon dioxide emissions has increased dramatically. Based on the estimation of agricultural carbon dioxide emissions from 1991 to 2018 in China, this paper uses EKC model to analyze economic growth and agricultural carbon dioxide emissions. The Kaya method is used to decompose the factors affecting agricultural carbon dioxide emissions. The experimental results show that there is a co-integration relationship between economic growth and the total intensity of agricultural carbon emissions, and between economic growth and the intensity of carbon emissions caused by five types of carbon sources: fertilizer, pesticide, agricultural film, agricultural diesel oil and tillage. Economic growth is the main driving factor of agricultural carbon dioxide emissions. In addition, technological progress has a strong role in promoting carbon emission reduction, but it has a certain randomness. However, the impact of energy consumption structure and population size on carbon emissions is not obvious.

## 1. Introduction

Global warming is an indisputable fact, which has seriously affected the survival environment and development of human beings. Apart from natural factors, climate warming is more caused by human activities, especially anthropogenic greenhouse gas emissions from the use of fossil fuels [1]. Carbon dioxide is one of the most important greenhouse gases, so carbon dioxide emission reduction is the most important thing to deal with climate change. In the past 30 years, China's economy has developed rapidly, and energy consumption has increased rapidly, which directly leads to a sharp increase in agricultural carbon dioxide emissions. With the concept of "low-carbon economy" proposed, how to reduce carbon dioxide emissions in agricultural production process and develop low-carbon agriculture is a matter of concern.

At present, some scholars at home and abroad have studied the relationship between agricultural carbon dioxide emissions and social economic growth. Ridzuan et al. [2] found that the relationship between carbon dioxide emissions and economic development is an inverted U, while Alkhathlan and Javid [3] believed that the relationship between them was a N-shaped

**Funding:** Long D. appreciates the Chongqing Social Science Planning Project - Research on Political Philosophy of the Construction of Human

Destiny Community (No. 19YBZX010). Meanwhile, Long D. also thanks the Western and Border Areas Project of 2018 Ministry of Education Humanities and Social Sciences Research Youth Fund - research on Shared Development Challenges in the context of the transformation of major social contradictions (No. 18XJC710008).

**Competing interests:** The authors have declared that no competing interests exist.

curve. Coondoo and Dinda [4] analyzed the relationship between agricultural carbon dioxide emissions and per capita income from the perspective of Granger causality, and found that different countries had different causality. Wu et al. [5] used LMDI method to study the changes of carbon emissions in China from 1980 to 2002 from the perspective of supply and demand. They believed that the scale of economic development, energy structure and energy intensity of energy demanders had driven the change of carbon emissions in China before 1996, and that the adjustment of industrial structure and the improvement of energy efficiency of energy demanders had played little role, while the improvement of energy efficiency of energy terminal utilization and conversion sectors from 1969 to 2000 was the main reason for the decrease of carbon emissions in China. Wang et al. [6] studied the change of carbon emissions in China from 1957 to 2000 using LMDI method. The results showed that China's carbon emissions decreased by 2466 million tons theoretically from 1957 to 2000, of which 95% was due to the decrease of energy intensity, only 1.6% and 3.2% were due to the adjustment of fossil energy structure and the utilization of renewable energy.

Economic development cannot be separated from the support role of agriculture. With the extensive application of fertilizers, pesticides and other agricultural materials, carbon emissions from agricultural production activities have gradually become an important part of global greenhouse gases. The close relationship with agriculture also determines the dialectical and unified relationship between economic development and agricultural carbon emissions [7–9]: on the one hand, the rapid economic development can promote the reduction of agricultural carbon emissions, such as the growth of agricultural environmental demand brought by economic development, the progress of pro-environment technology and the concept of sustainable economic development of farmers have become important drivers of agricultural carbon emission reduction; on the other hand, economic development has also contributed to the increase in agricultural carbon emissions. The driving effect of economic development on agricultural carbon emissions is more obvious, such as the increasing demand for agricultural products, rural surplus labor force and non-agricultural land use caused by economic development, which leads to a large number of carbon emissions. Based on this, the co-integration analysis of economic growth and agricultural carbon emission intensity data shows that the co-integration relationship between economic growth and agricultural carbon emissions, i.e. long-term equilibrium relationship, undoubtedly has important theoretical and practical reference significance for the development of agricultural environment and climate work in China.

The above research elaborates the relationship between agricultural carbon dioxide and the changes of socio-economic system from different angles, which has higher theoretical and practical value. With the deterioration of the environment and the increase in global temperature, scholars have increased their research on environmental protection and carbon emission reduction, especially on the green development of developed countries [10]. However, the current literature lacks the impact of socio-economic system changes on agricultural carbon dioxide emissions. This paper calculates agricultural carbon emissions and studies the impact of economic growth on agricultural carbon dioxide emissions based on the EKC model and Kaya method.

## 2. Background and methods

### 2.1 Environmental protection and development of socio-economic system

**2.1.1 Market economy, economic development and environmental protection.** Compared with planned economy, the vitality of market economy is more obvious and promotes economic growth. Because market economy is a competitive economy, without competition, people will have no pressure and motivation, no scientific development, technological

progress, efficiency improvement, and no great social development [11]. However, the disorderly market economy may cause damage to the ecological environment. Eco-environment and economic development are closely linked, and environment has the dual nature of promoting and restricting economic development. They are both opposite and unified. The purpose of economic development is to improve people's living standards and directly increase current interests. Only focusing on high-speed economic growth will inevitably sacrifice the ecological environment, which in turn will retard the speed of economic development. The Seventeenth National Congress of the Communist Party of China proposed that we should accelerate the transformation of the mode of economic development. The environmental protection cannot be separated from the transformation of the mode of economic development, that is, from the ecology of the mode of economic development. The so-called ecologization of economic development means to achieve economic development in a sustainable way [12], to adapt economic development to the ecological carrying capacity of nature, and to advocate resource conservation, environmental friendliness and ecological harmony between man and nature in economic development. Eco-economics or environmental economics is the subject of studying the ecology of economic development mode. In other words, we should reshape eco-economics or environmental economics according to the orientation of ecology of economic development mode.

**2.1.2 Socio-economic institutional change and macro-background of agricultural carbon dioxide emission reduction.**   The environmental problems caused by carbon emissions have attracted more and more attention. This is mainly because people are pursuing economic growth while seriously destroying the natural environment. In fact, this not only destroys the environment, but also has an impact on people's lives. Nowadays, global warming has become a fact. In order to control the continuous development of this situation, low-carbon economy emerged as a new form of economic development [13]. Therefore, to develop low-carbon economy, first of all, we should determine the way to develop low-carbon economy. The way is to adjust the economic structure, change the way of life and develop renewable energy technology. In addition, we should fully play the government's functions and improve the government's management level. Low-carbon economy is a new form of economic development designed to deal with greenhouse gas emissions on the surface. In fact, it also contains many contents. It is not only the main body of enterprise development but also the main mode of modern market economy development. To implement the low-carbon economy model, energy conservation and emission reduction must be carried out from many aspects. We should know that energy conservation and emission reduction are the basis of building a low-carbon civilization, which can promote the simultaneous development of environment and economic growth. Therefore, low-carbon economy is the only way for the sustainable development of the country. At the same time, low-carbon economy has become the guide for the sustainable development of the country [14–16]. It provides an operable path for sustainable development, including: low-carbon energy system, low-carbon industrial system, low-carbon technology system and so on. The relationship between carbon emissions and economic growth is the key to the sustainable development of low-carbon economy. Only by properly handling the relationship between carbon emissions and economic growth can we promote the sustainable development of the country.

## 2.2 Research methods and data sources

**2.2.1 Environmental Kuznets Curve(EKC) model for carbon emissions\.**   In order to analyze the relationship between social and economic growth and agricultural carbon dioxide emission reduction brought about by socio-economic system reform in the perspective of

environmental protection, this paper uses the simple regression equation (logarithmic form) of economic growth-environmental quality to analyze. Agricultural carbon dioxide emissions refer to greenhouse gas emissions directly or indirectly caused by chemical fertilizers, pesticides, energy consumption and land tillage in the process of agricultural production [17]. Generally speaking, agricultural carbon dioxide emissions mainly come from six aspects: firstly, the direct or indirect carbon emissions caused by the production and use of chemical fertilizers. Secondly, the carbon emissions caused by the production and use of pesticides. Thirdly, the carbon emissions caused by the production and degradation of agricultural film [18]. Agriculture films are also known as film plastic, which is mainly used to cover farmland to increase ground temperature, maintain quality soil moisture, promote seed germination, rapid growth of seedlings, and inhibit the growth of weeds. It produces carbon dioxide during production and degradation. and inhibit the growth of weeds. It produces carbon dioxide during production and degradation. Fourthly, the direct or indirect consumption of fossil fuels (mainly agricultural diesel oil) used by the agricultural machinery; fifthly, in agricultural production, ploughing has broken the soil organic carbon pool, and a large amount of organic carbon is lost to the air, resulting in carbon emissions; sixthly, carbon release from indirect consumption of fossil fuels by electricity during irrigation [19].

In the process of analysis, assuming that the influence of other factors except agricultural economic development on carbon dioxide emissions remains unchanged, the following equation is obtained by using the intercept term in the equation:

$$\ln y_i = a + b_1 \ln x_i + b_2 (\ln x_i)^2 + b_3 (\ln x_i)^3 + e_i \tag{1}$$

$$E = \sum E_i = \sum T_i \bullet \delta_i \tag{2}$$

In formula (1), $y_i$ represents the agricultural carbon dioxide emissions in the first year; $x_i$ represents the level of social and economic development in the second year (commonly expressed as GDP); $a$ is the intercept term, indicating the impact of other factors (population, technology, etc.); $e_i$ is the random error term; for different coefficient $b_i(i = 1,2,3)$, the meaning of the model is different. Specifically, when $b_1>0,b_2<0,b_3>0$ or $b_1<0,b_2>0,b_3<0$, it shows that there is a relationship between agricultural carbon dioxide and per capita GDP: N or inverted N curve; when $b_1<0,b_2>0,b_3 = 0$ or $b_1>0,b_2<0,b_3 = 0$, it shows that there is a relationship between agricultural carbon dioxide and per capita GDP: U or inverted U curve; when $b_3 = 0$, $b_2 = 0, b_1 \neq 0$, it shows that there is a monotonous linear relationship between agricultural carbon dioxide and per capita GDP.

In formula (2), $E$ is the total agricultural carbon dioxide emissions, $E_i$ is the carbon dioxide emissions of various carbon sources, $T_i$ is the amount of each carbon source, $\delta_i$ is the carbon emission coefficient of each carbon source. According to the relevant empirical data, the coefficients of agricultural carbon dioxide emission source are summarized as shown in Table 1.

**Table 1. Sources and coefficient of agricultural carbon dioxide emission.**

| carbon source | Coefficient |
|---|---|
| chemical fertilizer | 0.8956kg·kg$^{-1}$ |
| Pesticides | 4.9341 kg·kg$^{-1}$ |
| Agricultural film | 5.18 kg·kg$^{-1}$ |
| diesel oil | 0.5927 kg·kg$^{-1}$ |
| Ploughing | 312.6 kg·km$^{-2}$ |
| Agricultural irrigation | 20.476 kg·hm$^{-2}$ |

The coefficient of agricultural irrigation carbon dioxide emission is 25 kg·hm$^{-2}$. Considering that the demand for fossils for thermal power only leads to indirect carbon emissions, 25 kg·hm-2. are multiplied by the thermal coefficient (that is, the ratio of China's thermal power generation to the total power generation). According to the statistical yearbook data of China, the average thermal power coefficient is 0.819, therefore the actual coefficient of agricultural irrigation is 20.476 kg·hm$^{-2}$.

**2.2.2 Decomposition of carbon emissions change.** EKC model has theoretical limitations in revealing the relationship between social and economic growth and agricultural carbon dioxide emissions. It is difficult to fully explain the change of carbon emissions. Therefore, this paper further uses Kaya factor decomposition method to analyze [20], in order to quantitatively analyze the relative importance of various factors in the process of agricultural carbon emissions change. Kaya identity links economic growth, technological level and population size with carbon dioxide generated by human activities through a simple mathematical formula, which can be expressed as follows:

$$CO_2 = \frac{CO_2}{\text{ENG}} \times \frac{\text{ENG}}{\text{GDP}} \times \frac{\text{GDP}}{\text{POP}} \times \text{POP} \tag{3}$$

Where, $CO_2$, ENG, GDP and *POP* represent agricultural carbon dioxide, primary energy consumption, GDP and population respectively: CO2/ENG represents carbon dioxide emissions per unit of energy use, ENG/GDP represents energy consumption per unit of GDP, and GDP/POP represents per capita GDP.

According to the knowledge of calculus, the change rate of any parameter on the right side of formula (3) in any period of time will be considered to be approximately equal to the change rate of agricultural carbon dioxide emissions in the corresponding period [21], so formula (3) can be converted into:

$$d(\ln CO_2) = d\ln(\frac{CO_2}{\text{ENG}}) + d\ln(\frac{\text{ENG}}{\text{GDP}}) + d\ln(\frac{\text{GDP}}{\text{POP}}) + d\ln(POP) \tag{4}$$

The above decomposition model regards the change of agricultural carbon dioxide emissions as the result of four factors: energy emission coefficient (determined by energy consumption structure), energy intensity (determined by technology level), per capita GDP (determined by economic level) and population size. That is to say, the change of agricultural carbon dioxide emissions can be decomposed into four different effects [22]: energy effect, technology effect, economic and demographic effects. If the effect of agricultural carbon dioxide emissions caused by the change of a factor is positive, it means that the effect promotes carbon emissions, and its change value is the incremental impact of agricultural carbon emissions changes. Conversely, when the value is negative, the effect reduces carbon emissions.

**2.2.3 Data.** In the original data used in this paper, agricultural energy consumption comes from China Energy Statistics Yearbook, and agricultural population and GDP come from China Statistics Yearbook. Since all kinds of energy consumption are physical statistics, they must be converted into standard statistics. The specific conversion methods are as follows [23]: the standard conversion coefficient of raw coal is 0.7143 kgce/kg, that of coke is 0.9714 kgce/kg, crude oil and fuel oil is 1.4286 kgce/kg, gasoline and kerosene is 1.4714 kgce/kg, diesel is 1.4571 kgce/kg, natural gas is 133,000 kgce/10km$^3$, electricity power is 1229 kgce/10,000 kWh [24]. Since the statistical caliber of energy consumption is broad agriculture (including agriculture, forestry, animal husbandry and fishery), agricultural GDP are expressed by the output value of primary industry over the years (calculated at 1978 prices).

The data of fertilizers, pesticides, agricultural film, diesel oil, the sown area and irrigated area of crops come from China's Rural Statistical Yearbook, which is based on the actual usage

in that year; the data of tillage is based on the actual sown area of crops in that year [25], the data of agricultural irrigation is based on the actual algae area in that year.

## 2.3 Estimation of agricultural carbon dioxide emissions

According to the fourth assessment report of IPCC, agricultural carbon emissions mainly refer to carbon emissions directly or indirectly generated in the process of agricultural production, such as fertilizer, agricultural film, pesticides, agricultural machinery and agricultural tillage [26]. In this paper, the calculation of agricultural carbon emissions is based on the total energy consumption of various types of agriculture multiplied by their respective carbon emission coefficients, as shown in Formula (5):

$$C_t = \sum_{j=1}^{9} E_{jt}\eta_j \tag{5}$$

where $C_t$ is the total carbon dioxide emissions in the t-th year of agriculture, $E_{jt}$ is the j-th energy consumption in the t-th year, and $\eta_j$ is the carbon emission coefficient of the j-th energy. It can be seen that the determination of energy carbon emission coefficient has a great influence on the calculation of carbon dioxide emissions. According to Sun et al. [27], the carbon emission coefficients of various energy sources are as follows: coal's emission coefficient is 0.7476 tons of standard coal, gasoline's emission coefficient is 0.5532 tons of standard coal, diesel's emission coefficient is 0.5913 tons of standard coal, natural gas's emission coefficient is 0.4479 tons of standard coal, kerosene's emission coefficient is 0.3416 tons of standard coal, fuel oil's emission coefficient is 0.6176 tons of standard coal, crude oil's emission coefficient is 0.5854 tons of carbon/ton standard coal, power's emission coefficient is 2.2132 tons of carbon/ton standard coal and coke's emission coefficient is 0.1128 tons of carbon/ton standard coal. Formula (4) can be used to estimate the total amount of carbon dioxide emissions from 1991 in China. Furthermore, the per capita carbon dioxide emissions and intensity of carbon emissions can be calculated and the trend is as shown in the Fig 1.

Carbon emission intensity refers to the carbon dioxide emissions per unit of GDP growth. This index is mainly used to measure the relationship between economic growth and carbon emissions growth. If the carbon dioxide emissions per unit of GDP decrease while economic growth, then a low-carbon development model has been realized [28].

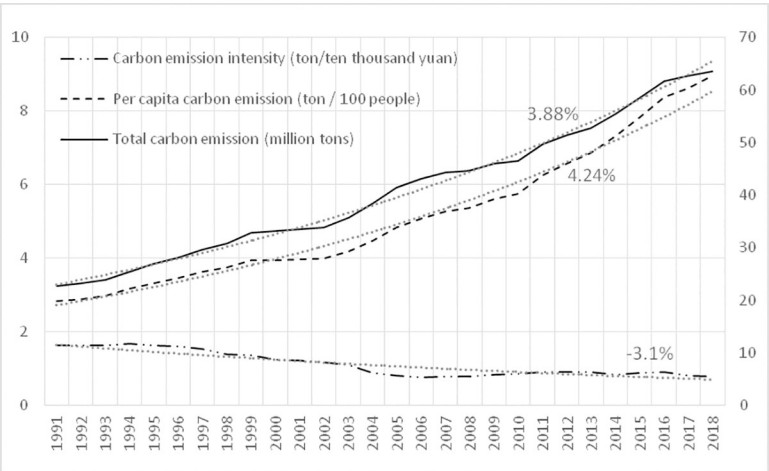

**Fig 1. Agricultural carbon dioxide emissions.**

From Fig 1, it can be seen that the total agricultural carbon dioxide emissions and per capita carbon dioxide emissions in China have an obvious growth trend during the research period. The total carbon dioxide emissions (see the right side ordinate axis of Fig 1) increased from 22.70 million tons in 1991 to 63.47 million tons in 2018, with an average annual growth rate of 3.88%; the per capita carbon dioxide emissions (see the left side ordinate axis of Fig 1) increased from 2.84 tons in 1981 to 8.95 tons in 2018, with an average annual growth rate of 4.24%. The intensity of carbon dioxide emissions (see the left side ordinate axis of Fig 1) in the study period showed a downward trend, from 163 tons of carbon per 10,000 yuan in 1981 to 0.79 tons of carbon per 10,000 yuan in 2018, with an average annual decline of 3.1%, indicating that China's agriculture is moving towards a healthy development of low-carbon mode. However, in the process of development, the intensity of carbon emissions has increased in some years, which shows that the impact of the adjustment of agricultural energy consumption structure and agricultural technology progress on carbon emissions is random [29].

## 3 Results

### 3.1 Cointegration test

**3.1.1 EKC relationship between carbon emissions and agricultural economic growth.** Cointegration test is the first step in regression analysis of time series. Cointegration is a method of modeling and theoretical analysis based on the combination of spatial structure and time dynamics developed on the basis of autoregression of time series vectors. Its meaning is that although each variable increases linearly and shows non-stationarity, one of their linear combinations is stationary, expressing a stable dynamic equilibrium relationship between two linear increments. Its birth makes up for many shortcomings of the traditional linear regression method based on least squares. In order to avoid the problem of "pseudo-regression" in the process of regression of non-stationary time series, unit root test of the analyzed sequence must be carried out before co-integration analysis. The results are shown in Table 2.

Unit root test results show that the original sequence $\ln y$ and $\ln x$ cannot reject the assumption that there is a unit root, so they are all unstable sequences. However, after first-order difference, both $\ln y$ and $\ln x$ become stationary sequences, which shows that they are first-order single-integer sequences, and cointegration analysis can be carried out. Johansen method is used to obtain the results of cointegration analysis as shown in Table 3.

The test results show that there is indeed a co-integration relationship between $\ln y$ and $\ln x$, that is, there is a long-term stable equilibrium relationship between them in the process of change, so there is no pseudo-regression problem in establishing the econometric equation with these two variables. According to formula (1), the relationship between agricultural

**Table 2. Sequence unit root test.**

| Variable | Test type (c, t, k) | ADF test value | P values | conclusion |
|---|---|---|---|---|
| lny | (c, t, 0) | -1.641 | 0.748 | Not smooth |
| dlny | (c, 0, 0) | -4.140 | 0.004*** | smooth |
| lnx | (c, t, 1) | -2.568 | 0.296 | Not smooth |
| dlnx | (c, 0, 0) | -2.923 | 0.057* | smooth |

Note: d stands for first-order difference; the test type (c, t, k) respectively indicates whether the unit root test contains constant term (c), time trend term (t) and lag order (k).

The number term and trend term were judged according to the sequence trend diagram, and the lag order was judged according to AIC and SC minimum criteria.

*, *** respectively represent rejecting the null hypothesis (the null hypothesis is the existence of a unit root) at the level of 10% and 1%.

**Table 3. ln$y$ and ln$x$ co-integration relation test.**

| The null hypothesis | The eigenvalue | The Trace statistic | 5% critical value | P values |
|---|---|---|---|---|
| Zero co-integration vectors | 0.475 | 33.342 | 29.797 | 0.019** |
| At most one cointegration vector | 0.237 | 6.762 | 15.495 | 0.606 |

Note

** means rejecting the null hypothesis at the 5% level.

carbon dioxide emissions and per capita GDP is analyzed, and the following regression equation is obtained:

$$\ln y = -94.346 + 51.285\ln x - 8.583(\ln x)^2 + 0.480(\ln x)^3 \tag{6}$$

In the regression equation, the t-test values of the regression coefficients are—6.468, 7.013, - 7.052 and 7.142, the $R^2$ value is 0.973, the F value is 278.73, and the DW value is 2.058, which shows that the overall fitting of the equation is good and the regression coefficient is significant. Because the coefficients $b_1 > 0$, $b_2 < 0$ and $b_3 > 0$, the relationship between curve of carbon dioxide and agricultural economic growth shows an obvious "N" trend, that is, along with agricultural economic growth, carbon emissions show a trend of rising first, then maintaining a certain level, and then rising again, which is different from the general inverted "U" type characteristics, indicating that there is a non-uniform relationship between carbon emissions and economic development in China's agriculture. This also proves that China's agricultural carbon emissions are volatile and inconsistent with economic growth. Therefore, in the following decomposition model, the changes of carbon emissions are further studied, and the effects of energy structure, economic development, technological level and population size on carbon emissions in different periods are preliminarily discussed.

**3.1.2 Cointegration test of agricultural carbon emissions and economic development.** In this paper, Engle-Granger Two-step Procedure is used to test the cointegration of carbon emission intensity, fertilizer, pesticide, agricultural film, agricultural diesel, irrigation, tillage carbon emission intensity and per capita GDP time series. In order to prevent the occurrence of Spurious Regression, Unit Root Test is used to test the eight time series. The eight variables are as follows: gdp is the per capita GDP, tci represents the intensity of total carbon emissions, cf is chemical fertilizer, pes is the pesticides, af is the agricultural film, do is diesel oil, plo means ploughing, irr is irrigation, respectively. And t represents time series. In order to better reflect the relationship between the intensity of carbon dioxide emissions and economic growth, we take logarithms of carbon dioxide emissions and real per capita GDP. The capital "L" stands for the logarithmic value of the variable. The Augmented Dickey-Fuller test (ADFest) is used to test the stationarity of the three models. It is a t-test for the following three models in order:

$$\Delta y_t = \gamma y_{t-i} + \sum_{i=1}^{p} \beta_i \Delta y_{t-i} + \varepsilon_t$$

$$\Delta y_t = \alpha + \gamma y_{t-i} + \sum_{i=1}^{p} \beta_i \Delta y_{t-i} + \varepsilon_t \tag{7}$$

$$\Delta y_t = \alpha + \gamma y_{t-i} + \beta_0 tr \sum_{i=1}^{p} \beta_i \Delta y_{t-i} + \varepsilon_t$$

The models mentioned above correspond to three regression forms, i.e. there are not intercept and time trend, only intercepts are included (random variable y, constant term $\alpha$, cointegration vector $\beta_i$, influence parameter $\beta_0$), there are intercept and time trend. Eviews 5.0 is used to automatically determine the lag order according to AIC criterion. The ADF test results are shown in Table 4.

The original Lgdp and Lcf sequences are stable sequences. The Ltci, Lpes, Lirr, and Lplo are first-order monolithic sequences after first-order difference. Laf and Ldo are second-order monolithic sequences because they can achieve the stationarity requirement through second-order difference. In order to verify the possible cointegration relationship between time series GDP and carbon emission intensity indicators in agricultural production activities caused by chemical fertilizers, pesticides, agricultural film, etc., the long-term trend equation of carbon emission intensity of each carbon source ($\ln y_t = \alpha + \beta \ln r g d p_t + \varepsilon_t$) is obtained (see in Table 5). The results are shown in Table 6, and the unit root test of the residual term $\varepsilon_t$ is carried out in the Fig 2.

Because Lirr-Lgdp regression test fails, it shows that Lgdp cannot explain the dependent variables linearly. Therefore, the relationship between them will not be considered in the next analysis.

According to Granger's cointegration theorem, there is a cointegration relationship between economic growth and total carbon emission intensity, fertilizer, pesticide, agricultural film, diesel oil and tillage. The cointegration vectors are (4.3934, 0.2035), (4.4195, 0.1474), (2.1424, 0.2011), (0.2773, 0.4234), (1.2968, 0.3270), (2.2073, -0.0900), respectively. The corresponding error correction term ECM is as follows:

Carbon emission intensity—economic growth ECM:

$$ecm(Ltci) = Ltci_t - 0.3934 - 0.2035 Lgdp_t$$

Carbon emission intensity of fertilizer use- economic growth ECM:

$$ecm(Lcf) = Lcf_t - 4.4195 - 0.1474 Lgdp_t$$

Carbon emission intensity of pesticide use- economic growth ECM:

$$ecm(Lpes) = Lpes_t - 2.1424 - 0.2011 Lgdp_t$$

Carbon emission intensity of agricultural film use—economic growth ECM:

$$ecm(Laf) = Laf_t - 0.2773 - 0.4234 Lgdp_t$$

Carbon emission intensity of Diesel Use—economic growth ECM:

$$ecm(Ldo) = Ldo_t - 1.2968 - 0.3270 Lgdp_t$$

Carbon emission intensity of ploughing-economic growth ECM:

$$ecm(Lplo) = Lplo_t - 2.2073 - 0.0900 Lgdp_t$$

## 3.2 Kaya decomposition results of carbon emissions

In order to analyze the different impacts of various factors on the changes of agricultural carbon emissions in China, the agricultural carbon emissions from 1991 to 2018 are decomposed year by year according to formula (4). The results are shown in Fig 2.

The annual average of energy effect is a small positive number (0.327%). The small energy effect shows that the overall impact of the adjustment of agricultural energy structure on

**Table 4. ADF unit root test results of agricultural carbon emission intensity and economic growth variables.**

| variable | Inspection form(c,t,k) | ADF test statistics | 10% critical value | Result |
|---|---|---|---|---|
| Lgdp | (c,t,3) | -3.684564 | -3.38833 | stable |
| Ltci | (c,t,1) | -2.015344 | -3.342253 | unstable |
| Δ(Ltci) | (c,t,1) | -5.328762 | -3.362984 | stable |
| Lcf | (c,t,2) | -14.89312 | -3.362984 | stable |
| Lpes | (c,t,3) | -0.56482 | -3.38833 | unstable |
| Δ(Lpes) | (c,t,2) | -8.291101 | -3.38833 | stable |
| Laf | (c,t,3) | -0.881726 | -3.38833 | unstable |
| Δ(Laf) | (c,t,3) | -2.022575 | -3.42003 | unstable |
| Δ(Laf,2) | (c,t,0) | -4.323096 | -3.362984 | stable |
| Ldo | (c,t,0) | -2.296447 | -3.324976 | unstable |
| Δ(Ldo) | (c,t,3) | -2.01862 | -3.42003 | unstable |
| Δ(Ldo,2) | (c,t,0) | -6.568117 | -3.362984 | stable |
| Lirr | (c,t,0) | -1.854383 | -3.324976 | unstable |
| Δ(Lirr) | (c,t,1) | -3.61596 | -3.362984 | stable |
| Lplo | (c,t,0) | -1.894755 | -3.324976 | unstable |
| Δ(Lplo) | (c,t,0) | -4.281394 | -3.342253 | stable |

Note: (c, t, k) Indicates whether there is a constant intercept term, a time trend term and an optimal lag order K in the ADF test. _is a first-order difference and_(2) is a second-order difference.

agricultural carbon dioxide emissions is not obvious. In fact, China's agricultural energy consumption structure has not changed significantly over the years. The main types of energy consumption include diesel, coal and electricity, of which diesel accounts for 39.50% of the total energy consumption, coal for 35.18%, electricity for 9.96%, and all other types of energy account for only 15.36%, and this consumption structure will not change significantly in the long future. The energy effect shows that the change of energy structure has not reduced carbon emissions, but has played a role in promoting it to a certain extent. It further shows that the structure of agricultural energy consumption in China has not been optimized, but has slightly deteriorated.

From the perspective of technology effect, the annual average value is significantly negative (-3.814%), which indicates that the progress of agricultural production technology has greatly alleviated the emission of agricultural carbon dioxide, and further shows that with the progress of agricultural technology, it is possible to develop low-carbon agriculture in China. It is worth noting that the technological effects fluctuate greatly over the years and do not show certain regularity, which indicates that the effect of agricultural technological progress on agricultural carbon dioxide emission reduction is random. For example, from 1991 to 2001, the effect of technology has been negative. Especially during the period of the Eighth Five-Year Plan

**Table 5. Logarithmic linear regression of agricultural carbon emission intensity and economic growth.**

| regression equation | Ltci | Lcf | Lpes | Laf | Ldo | Lirr | Lplo |
|---|---|---|---|---|---|---|---|
| Intercept term | 4.3934 | 4.4195 | 2.1424 | 0.2773 | 1.2968 | 2.3639 | 2.2073 |
| Lgdp | 0.2035 | 0.1474 | 0.2011 | 0.4234 | 0.327 | -0.0169 | -0.09 |
| $R^2$ | 0.6648 | 0.5145 | 0.6767 | 0.9087 | 0.7417 | 0.0105 | 0.2073 |
| F value | 27.7662 | 14.8336 | 29.3052 | 139.3335 | 40.2007 | 0.1486 | 4.7828 |
| Prob(F-statistic) | 0.0001 | 0.0018 | 0.0001 | 0 | 0 | 0.7057 | 0.0462 |
| DW value | 0.8917 | 0.8303 | 1.1936 | 0.9268 | 1.0587 | 0.6995 | 0.7836 |

**Table 6. Result of ADF unit root test of residuals.**

| variable | (c, t, k) | ADF test statistics | 10% critical value | Result |
|----------|-----------|---------------------|---------------------|--------|
| e(Ltci) | (c, t, 2) | -13.16318 | -3.362984 | stable |
| e(Lcf) | (c, t, 2) | -18.46234 | -3.362984 | stable |
| e(Lpes) | (c, t, 2) | -7.11588 | -3.362984 | stable |
| e(Laf) | (c, t, 1) | -3.538049 | -3.342253 | stable |
| e(Ldo) | (c, t, 2) | -5.537185 | -3.362984 | stable |
| e(Lplo) | (c, t, 2) | -23.72241 | -3.362984 | stable |

(1991–1995), the effect of technological progress on agricultural carbon emission reduction is very obvious, while from 1996 to 2002, the effect of technology has been positive continuously, indicating that technological progress has not alleviated agricultural carbon dioxide emissions.

From the perspective of economic effect, except one year is negative, the other years are significantly positive (the annual average is 7.760%). This shows that the rapid development of agricultural economy has greatly promoted agricultural carbon dioxide emissions, and economic growth is the main driving factor of agricultural carbon emissions. It is also noteworthy that the effect of the economy on carbon emissions is also volatile, with some years having large effects and some years having smaller effects. For example, during the five-year period of the Eighth Five-Year Plan, the economic effect is very great, because the five-year period is a period of rapid development of Chinese economy, and the rapid development of the agricultural economy has also led to the emission of agricultural carbon dioxide; the economic effect has declined dramatically, which may be due to the measures taken by China to eliminate and close a number of small agricultural enterprises with backward technology, waste of resources and the financial crisis. From 2000 to 2008, the economic effect rose again in fluctuation. This may be due to the macro-policy of expanding domestic demand and increasing investment, which led to a large number of high energy consumption and repetitive agricultural infrastructure projects blindly launched, resulting in the contribution of agricultural economic development to carbon emissions kept high.

From the point of view of population effect, the annual fluctuation is not significant (the minimum is—1.793%, the maximum is 1.164%), which indicates that the change of agricultural population size has no obvious impact on agricultural carbon dioxide emissions. It is worth noting that the population effect can be divided into two distinct stages. On the one hand, it is caused by the changes in the number of agricultural population. On the other hand, it is possible that with the development of society and economy, the rise of energy prices and the gradual strengthening of farmers' low-carbon consciousness, so more attention should be paid to the use of energy saving in production and life, and agricultural carbon dioxide emissions has been reduced to a certain extent.

## 4. Discussion

In view of the needs of environmental protection, the following suggestions are put forward for carbon dioxide emission reduction:

First, it should give full play to the technological effect and find out reasonable emission reduction measures to reduce the intensity of carbon dioxide emissions. This requires us to develop clean energy technology (electricity, wind, solar energy) to replace non-clean energy (coal, oil). The total carbon dioxide emissions from the whole economy will be reduced by 1.5 units for each additional unit of electricity input in the two major energy sectors—the power sector and the natural gas sector. This requires the increase of China's natural energy

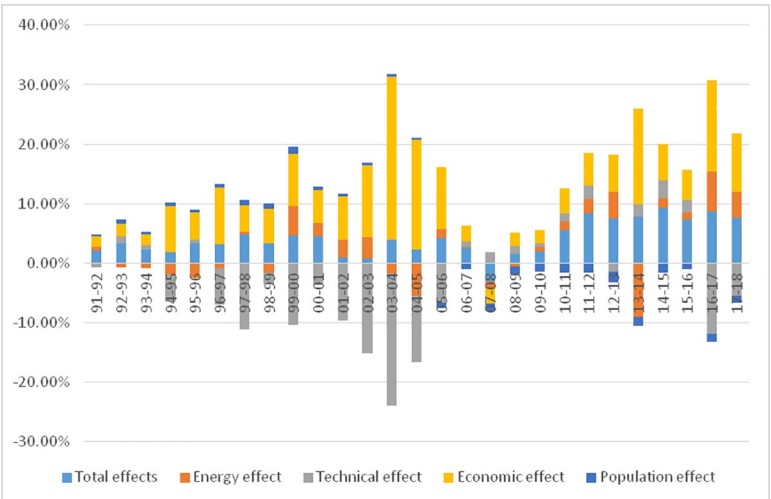

**Fig 2. The contribution rate of each effect in the change of agricultural carbon emission in China.**

infrastructure construction projects, such as electricity, wind energy, to meet the requirements of limiting carbon dioxide emissions while developing the economy. On the other hand, technologies to reduce the emission intensity of non-clean energy should be developed. According to statistics, the utilization efficiency of raw coal and oil in China is far lower than that in developed countries. Energy waste caused by backward technology is an important reason for the rising intensity of carbon dioxide emissions, which requires the government to increase financial and technical support.

Secondly, effective measures should be taken to improve the utilization rate of agricultural resources such as water conservancy and land, and reduce energy consumption per unit of agricultural output as much as possible. Adjusting and optimizing agricultural structure are the objective requirement for agriculture to enter a new stage and enhance its development capacity, and is also the inevitable product of agricultural market reform and development. Therefore, we should speed up the pace of adjustment, vigorously implement and develop agricultural circular economy models, such as the recycling of resources and energy linked by crop straw and the recycling of agricultural by-products as raw materials, and further optimize and renew agricultural machinery and equipment and agricultural technology in order to reduce carbon dioxide emissions caused by the use of agricultural machinery and equipment; relevant departments should establish and improve the legal and regulatory system of resource utilization and emission reduction and energy conservation, so as to provide reliable legal protection for agricultural emission reduction. The government should strengthen investment in low-carbon agriculture and provide corresponding policy support to ensure the smooth implementation of agricultural emission reduction.

## 5. Conclusions

This paper studies the impact of socio-economic system changes on agricultural carbon dioxide emission reduction from the perspective of environmental protection. The economic growth brought about by the change of economic system has promoted the growth of carbon dioxide emissions from fertilizers, pesticides, agricultural film, agricultural diesel, irrigation and tillage to varying degrees. It shows that China's economic growth, especially agricultural growth mode, belongs to resources and energy consumption-oriented. The change of

agricultural energy consumption structure has no obvious effect on carbon dioxide emission reduction, and even exacerbates carbon dioxide emission to a certain extent. This shows that China's agriculture still has strong dependence on traditional fossil energy (especially high-emission diesel oil), which is very unfavorable to agricultural carbon dioxide emission reduction. Speeding up the development and utilization of renewable energy and promoting the construction of renewable energy laws and regulations will be an important strategic choice for China's agricultural carbon dioxide emission reduction in the future.

## Supporting information

**S1 Data.**
(XLSX)

## Author Contributions

**Writing – original draft:** Deng Jie Long.

**Writing – review & editing:** Li Tang.

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
