## [Decision Letter · Decision Letter 0]

9 Feb 2021

PONE-D-20-35559

The Impact of Socio-economic Institutional Change on Agricultural Carbon Dioxide Emission Reduction from the Perspective of Ecological Civilization Construction

PLOS ONE

Dear Dr. Long,

Thank you for submitting your manuscript to PLOS ONE. After careful consideration, we feel that it has merit but does not fully meet PLOS ONE’s publication criteria as it currently stands. Therefore, we invite you to submit a revised version of the manuscript that addresses the points raised during the review process.

We look forward to receiving your revised manuscript.

Kind regards,

Zhihua Zhang

Academic Editor

PLOS ONE

"Chongqing Social Science Planning Project - Research on Political Philosophy of the Construction

of Human Destiny Community (No. 19YBZX010); This paper is a phase achievement of the

Western and Border Areas Project of 2018 Ministry of Education Humanities and Social Sciences

Research Youth Fund - research on Shared Development Challenges in the context of the

transformation of major social contradictions (No. 18XJC710008)."

"All relevant data are within the manuscript and its Supporting Information files."

4. We suggest you thoroughly copyedit your manuscript for language usage, spelling, and grammar. If you do not know anyone who can help you do this, you may wish to consider employing a professional scientific editing service.  

Reviewers' comments:

Reviewer's Responses to Questions

**Comments to the Author**

1. Is the manuscript technically sound, and do the data support the conclusions?

Reviewer #1: Partly

Reviewer #2: Yes

2. Has the statistical analysis been performed appropriately and rigorously? 

Reviewer #1: I Don't Know

Reviewer #2: Yes

3. Have the authors made all data underlying the findings in their manuscript fully available?

Reviewer #1: Yes

Reviewer #2: Yes

4. Is the manuscript presented in an intelligible fashion and written in standard English?

Reviewer #1: No

Reviewer #2: Yes

5. Review Comments to the Author

Reviewer #1: I have provided a review and it is attached. I do not find that the manuscript reports new findings or insights. The manuscript is also not easy to follow and some of the terminology used is either obscure or not defined.

Reviewer #2: This paper has studied the impact of socio-economic system changes on agricultural carbon dioxide emission reduction from the perspective of ecological civilization construction. The work of this paper is clear. However, there are some problems to be further improved as well:

Be sure that the explanations of research methods are detailed and correct. For example:

The meaning of constants b_i for the model in formula (1) remains to be checked.

Formula (6) is incomplete.

The names of variables in 3.1(2) are inconsistent during the manuscript

It is noted that your manuscript needs careful editing which contains many mistakes. The units in table 1 and its description in the paper “the actual coefficient of agricultural irrigation is kg·hm-2 ” remain to be checked, for example.

6. PLOS authors have the option to publish the peer review history of their article (what does this mean?). If published, this will include your full peer review and any attached files.

Reviewer #1: No

Reviewer #2: No

---

## [Author Response · Author response to Decision Letter 0]

9 Apr 2021

Response to Reviewers’ Comments

Manuscript # PONE-D-20-35559

We appreciate the editor giving us the opportunity to revise the paper. We also thank the reviewers for their feedback and helpful suggestions that have substantially improved the quality of our paper. All the authors have seriously discussed about all these comments. According to the reviewers’ comments, we have tried best to modify our manuscript to meet with the requirements of your journal. Our answers are provided in blue and the text revisions are highlighted.

Reviewer #1: 

The manuscript by Long and Tang aims to use the “EKC model to analyze economic growth and agricultural carbon dioxide emissions in the perspective of ecological civilization construction”. Their experimental results show that the relationships between economic growth and the total intensity of agricultural carbon emissions, and between economic growth and the intensity of carbon emissions caused by five types of carbon sources: fertilizer, pesticide, agricultural film, agricultural diesel oil and tillage are integrated. In addition, economic growth was identified as the main driving factor of agricultural carbon dioxide emissions. I found this manuscript very difficult to follow. At the outstart I was confused by the term “ecological civilization construction” and still I am not clear on what this means. Furthermore, I found that the findings of the work were not surprising and couldn’t help but think throughout reading the manuscript, that we already know much of what the authors are claiming to reveal. Which brings me to what is significant in this research and manuscript. I also found that when I referred to the literature that was cited, in many cases I could not fathom how the citation was related to the point being made (e.g., #7). Many of the references seem to be either obscure or perhaps incorrect. Overall, then, I do not recommend that this manuscript be published and at a minimum it requires a major revision.

Answer: Thanks for your enlightening comment. Firstly, the ecological civilization construction was proposed by Chinese leaders, which refers to the establishment of the concept of respecting nature, conforming to nature, and protecting nature to cope with the tightening of resource constraints and serious environmental pollution. Considering that this journal is an international academic journal, we have changed “ecological civilization construction” and “construction of ecological civilization” to “environmental protection” and revised the paper title to “The Impact of Socio-economic Institutional Change on Agricultural Carbon Dioxide Emission Reduction in China”. Secondly, this manuscript revised some statements to keep objective and truthful, such as changing “In addition, the current literature data are relatively old, and the existing research conclusions are quite different [7]” to “With the deterioration of the environment and the increase in global temperature, scholars have increased their research on environmental protection and carbon emission reduction, especially on the green development of developed countries [7]”. Furthermore, we carefully checked the consistency of the citations and manuscript content, and replaced some inappropriate citations, such as citation [7]. Finally, we carefully checked the content of the full text and corrected some errors to make this manuscript meet the publishing requirements as much as possible.

A few small comments:

1. What is an agricultural film (p. 10) and how does its use produce CO2 emissions?

Answer: Thank you for your comments. Agriculture films are also known as film plastic, which refers to the use of plastic materials in various agricultural applications. It is mainly used to cover farmland to increase ground temperature, maintain quality soil moisture, promote seed germination, rapid growth of seedlings, and inhibit the growth of weeds. In the production of agricultural films, carbon dioxide is inevitably produced. In addition, since the main component of the agricultural film is polyethylene (nCH2), carbon dioxide will also be produced with degradation of agricultural film. We have made a footnote to the agricultural film for specific explanations.

2. Why not present the data in Table 2 graphically – I think it would be much more effective and communicate your point better.

Answer: Thank you for your suggestions. We have visualized Table 2 as Figure 1, from which we can intuitively see the change trend. And we added the trend lines and average annual growth rate of the three indicators in Fig.1. The Fig.1 is as follows:

3. Page 15 – is it necessary to spell out all the logs of CO2 emissions and real per capita GDP?

Answer: Thank you for your suggestions. We have deleted the specific explanation of all the logarithm of CO2 emissions and real per capita GDP. The revised content is as follows:

The eight variables are as follows: gdp is the per capita GDP, tci represents the intensity of total carbon emissions, cf is chemical fertilizer, pes is the pesticides, af is the agricutural film, do is diesel oil, plo means ploughing, irr is irrigation, respectively. And t represents time series. In order to better reflect the relationship between the intensity of carbon dioxide emissions and economic growth, we take logarithms of carbon dioxide emissions and real per capita GDP. The capital "L" stands for the logarithmic value of the variable.

4. Table 8 – is there a more compelling way to present these date and to identify trends –is there a trend?.

Answer: Thank you for your comments. This table mainly wants to show the contribution rate of each effect in the changes in China's agricultural carbon emissions. Because it is the decomposition of the change value of the variable, the contribution rate of each effect fluctuates greatly every year, and there is no significant trend of change. Following your suggestion, we replaced Table 8 with a histogram, which is more intuitive and obvious. The histogram is as follows:

Reviewer #2: 

This paper has studied the impact of socio-economic system changes on agricultural carbon dioxide emission reduction from the perspective of environmental protection. The work of this paper is clear. However, there are some problems to be further improved as well:

Be sure that the explanations of research methods are detailed and correct. For example: 

1) The meaning of constants b_i for the model in formula (1) remains to be checked.

Answer: Thank you for your suggestions. We have modified expression of the meaning bi. The revised content is as follows:

for different coefficient, the meaning of the model is different. Specifically, when or , it shows that there is a relationship between agricultural carbon dioxide and per capita GDP: N or inverted N curve; when or , it shows that there is a relationship between agricultural carbon dioxide and per capita GDP: U or inverted U curve; when , it shows that there is a monotonous linear relationship between agricultural carbon dioxide and per capita GDP.

2) Formula (6) is incomplete.

Answer: Thank you for pointing this out. We have completed formula (6). The modified formula (6) are as follows:

3) The names of variables in 3.1(2) are inconsistent during the manuscript

Answer: Thank you for pointing out the error. We have modified and checked so that the variable names are uniform are inconsistent in the paper. The adjusted variable names are as follows: tci, cf, pes, do, plo and irr. 

4) It is noted that your manuscript needs careful editing which contains many mistakes. The units in table 1 and its description in the paper “the actual coefficient of agricultural irrigation is kg·hm-2 ” remain to be checked, for example.

Answer: Thank you for your suggestions. We have corrected the error and carefully checked the full paper to keep the paper rigorous and correct. 

See all figures and tables in attached file.

---

## [Decision Letter · Decision Letter 1]

4 May 2021

The Impact of Socio-economic Institutional Change on Agricultural Carbon Dioxide Emission Reduction Based in China

PONE-D-20-35559R1

Dear Dr. Long,

We’re pleased to inform you that your manuscript has been judged scientifically suitable for publication and will be formally accepted for publication once it meets all outstanding technical requirements.

Kind regards,

Zhihua Zhang

Academic Editor

PLOS ONE

Additional Editor Comments (optional):

Reviewers' comments:

Reviewer's Responses to Questions

**Comments to the Author**

1. If the authors have adequately addressed your comments raised in a previous round of review and you feel that this manuscript is now acceptable for publication, you may indicate that here to bypass the “Comments to the Author” section, enter your conflict of interest statement in the “Confidential to Editor” section, and submit your "Accept" recommendation.

Reviewer #2: All comments have been addressed

2. Is the manuscript technically sound, and do the data support the conclusions?

Reviewer #2: Yes

3. Has the statistical analysis been performed appropriately and rigorously? 

Reviewer #2: Yes

4. Have the authors made all data underlying the findings in their manuscript fully available?

Reviewer #2: Yes

5. Is the manuscript presented in an intelligible fashion and written in standard English?

Reviewer #2: Yes

6. Review Comments to the Author

Reviewer #2: (No Response)

7. PLOS authors have the option to publish the peer review history of their article (what does this mean?). If published, this will include your full peer review and any attached files.

Reviewer #2: No

---

## [Editor Report · Acceptance letter]

10 May 2021

PONE-D-20-35559R1 

The Impact of Socio-economic Institutional Change on Agricultural Carbon Dioxide Emission Reduction in China 

Dear Dr. Long:

I'm pleased to inform you that your manuscript has been deemed suitable for publication in PLOS ONE. Congratulations! Your manuscript is now with our production department. 

Kind regards, 

on behalf of

Dr. Zhihua Zhang 

Academic Editor

PLOS ONE